# Demographics and Trends for Inbound Medical Tourism in Germany for Orthopedic Patients before and during the COVID-19 Pandemic

**DOI:** 10.3390/ijerph20021209

**Published:** 2023-01-10

**Authors:** Tizian Heinz, Annette Eidmann, Axel Jakuscheit, Tino Laux, Maximilian Rudert, Ioannis Stratos

**Affiliations:** 1Department of Orthopaedic Surgery, Julius-Maximilians University Wuerzburg, Koenig-Ludwig-Haus, Brettreichstrasse 11, 97074 Wuerzburg, Germany; 2Distance and Independent Studies Center, Technical University Kaiserslautern, Erwin-Schroedinger-Straße 57, 67663 Kaiserslautern, Germany

**Keywords:** inbound medical tourism, orthopedic surgery, Germany, COVID-19, pandemic

## Abstract

Medical tourism is a rapidly growing sector of economic growth and diversification. However, data on the demographics and characteristics of the traveling patients are sparse. In this study, we analyzed the common demographic properties and characteristics of the inbound medical tourists seeking orthopedic medical care in Germany for the years 2010 to 2019 compared to a domestic group. At the same time, we examined how the COVID-19 pandemic outbreak of 2020 changed the field of medical tourism in Germany. Calculations were performed using administrative hospital data provided by the Federal Statistical Department of Germany. Data were analyzed from the years 2010 to 2020. A total of six elective orthopedic surgery codes (bone biopsy, knee arthroplasty, foot surgery, osteotomy, hardware removal, and arthrodesis) were identified as key service indicators for medical tourism and further analyzed. Factors including residence, sex, year, and type of elective surgery were modeled using linear regression analysis. Age and sex distributions were compared between patients living inside Germany (DE) or outside Germany (non-DE). Between 2010 and 2020, 6,261,801 orthopedic procedures were coded for the DE group and 27,420 key procedures were identified for the non-DE group. Medical tourists were predominantly male and significantly younger than the domestic population. The linear regression analysis of the OPS codes over the past years showed a significantly different slope between the DE and non-DE groups only for the OPS code “hardware removal”. With the COVID-19 pandemic, an overall decline in performed orthopedic procedures was observed for the non-DE and the DE group. A significant reduction below the 95% prediction bands for the year 2020 could be shown for hardware removal and foot surgery (for DE), and for hardware removal, knee arthroplasty, foot surgery, and osteotomy (for non-DE). This study is the first to quantify inbound medical tourism in elective orthopedic surgery in Germany. The COVID-19 pandemic negatively affected many—but not all—areas of orthopedic surgery. It has to be seen how this negative trend will develop in the future.

## 1. Introduction

Medical tourism (MT) is defined as the process where patients travel outside the geographic borders of their home region or home country to obtain medical care. The driving force for medical travel includes but is not limited to the affordability of medical procedures, easier access to care and health, as well as more developed and specialized medical care in foreign regions [1]. Many authors subcategorize MT into domestic, inbound, and outbound. Domestic MT refers to patients who seek medical treatment outside their hometown or home region but stay within the geographic borders of their country. Inbound MT includes patients that cross borders into a specific foreign country to receive medical care, whereas outbound medical tourists leave their home country for medical care abroad [2]. As a result, a nation can treat patients traveling from other countries (inbound medical tourism), while at the same time, the citizens of this nation can seek medical treatment abroad (outbound medical tourism). 

Patients are increasingly transforming into informed consumers, who choose their own providers out of a broad market place that is not limited to geographical borders. MT has also become a prominent phenomenon in the European Union (EU). The financial and economic impact of EU citizens seeking medical care in Germnay has generated an estimated economic volume of about 200 million EUR per year, thereby being one of the highest of all EU member states, not yet including international patients from outside the EU seeking medical care in Germany [3]. This demonstrates that MT, though a relatively young branch of international trade, has advanced into a billion-dollar market during the last decades, attracting the interest of the scientific community as a potential sector of economic growth and diversification [4]. MT is already contributing 1% (aproximatelly 10 billion EUR) to the publicly financed EU health market [5]. Germany is actively participating in this market. From a global point of view, Germany is ranked as the 12th most attractive MT destination among 46 countries [6]. Schmerler also characterizes Germany as one of the leading destinations for inbound medical tourists among the U.K., U.S., Russia, Australia, and the UAE [7]. Additionally, Germany has a good reputation among medical travelers, making it one of the most popular destinations of medical care seekers [4]. Against the background of Europe being a popular destination for MT, it is of outmost importance to analyze and screen inbound patient flows for the establishment of a framework towards controlled cross-border healthcare, thereby enhancing the related benefits of MT, while at the same time recognizing and minimizing potentially adverse effects. 

Predominantly, surgical departments are popular destinations for medical tourists [8,9]. Specifically, orthopedic surgery departments have a substantial financial benefit from inbound MT. According to Lunt et al. [8], orthopedic surgical treatments are frequent medical services for inbound medical tourism. Moreover, procedures such as elective hip and knee replacement, arthroscopy, and spinal surgery are commonly pursued by medical travelers [10]. During the COVID-19 pandemic, non-essential travel to Germany was restricted by the government during the year 2020. Recent reports show that medical tourism in Germany declined compared to previous years and further reduced after the year 2020 [11]. 

The intention of this study was twofold: Firstly, it was the aim of this analysis to investigate how travel and healthcare-related organizational restrictions in Germany due to the COVID-19 panedmic would translate to a visibly altered framework of MT. Secondly, the trend and development of MT in the orthopedic field in Germany was to be investigated. Therefore, the demographics and frequency of selected orthopedic services provided to medical tourists were analyzed for the years 2010–2020. 

## 2. Materials and Methods

### 2.1. Source Data

Calculations were performed using the case-based hospital statistics dataset 23141-0103 from the Federal Statistical Department of Germany (title: Operations and procedures for inpatients: Germany, years, sex, age groups, patient’s place of residence, operations and procedures; https://www-genesis.destatis.de, accessed on 27 September 2022). The data were retrieved on 30 June 2022. The dataset 23141-0103 contains every surgical procedure coded in any hospital in Germany, as well as patient demographic data (sex, age, and permanent residence). Analysis focused from 2010 to 2020. The search also included the year 2020, which marked the beginning of the COVID-19 outbreak’s global travel restrictions. 

### 2.2. OPS Codes and Surgical Procedure Identification

Each surgical procedure coded in the 23141-0103 dataset is based on the “Operation and Procedure Classification System” (OPS). The OPS is currently the official coding system for medical procedures in Germany. The monohierarchical classifications of the OPS System organize medical procedures into classes of different hierarchical levels: chapters are divided into groups, groups into categories; categories usually have subcategories. Due to this detailed structure, the entire OPS catalog contains 1685 OPS codes in the four-digit hierarchy level. Out of these 1685 codes, we identified six OPS codes that are clearly associated with elective orthopedic surgical procedures. These “elective orthopedic OPS codes” were bone biopsy by incision (OPS-1-503; bone biopsy), implantation of endoprosthesis of knee joint (OPS-5-822; knee arthroplasty), operations on metatarsals and phalanges of the foot (OPS-5-788; foot surgery), osteotomy and corrective osteotomy (OPS-5-781; osteotomy) and removal of osteosynthesis material (OPS-5-787; hardware removal), and arthrodesis (OPS-5-808; arthrodesis).

### 2.3. Data Processing

A table containing grouped demographic data and OPS codes was generated from the 23141-0103 dataset. The table contained the following data: elective orthopedic OPS codes (with 6 subcategories: “OPS-1-503”, “OPS-5-822”, “OPS-5-788”, ”OPS-5-787”, “OPS-5-808”), age (with 22 subcategories: “under 1”, “1–5”, “5–10”,…,”85–90”, “90–95”, “over 95”), sex (two subcategories: “male” and “female”), year of acquisition (11 subcategories: “2010”, “2011”, …, “2018”, “2019” and “2020”), and permanent residence (18 subcategories: “16 German states”, “foreign”, and “unknown”). All tabled data were unpivoted using R (RStudio v.1.3.1093; Boston, United States). The category “permanent residence” was rearranged into 2 subcategories: “Germany” (DE; 8,195,795 OPS codes) and “abroad” (non-DE; 38,624 OPS codes). Data from “unknown” residence (14,478 OPS codes) were excluded from the analysis. Furthermore, the age category was rearranged into 4 subcategories: “0–17”, “18–39”, “40–64”, and “over 65”. Data summarization was performed using the visual analytics software Tableau Desktop (Tableau Software, v. 2021, Seattle, WA, USA).

### 2.4. Statistical Analysis

A linear regression model was used to describe and model both the effects of sex and the number of orthopedic OPS-coded procedures for the non-De and the De groups over time. The slopes of the correlation lines were compared, and *p*-values were calculated to test the null hypothesis. To elucidate if the observed data from the year 2020 were within the statistical expectation, a regression analysis was performed for the years 2010–2019. The year 2020 was excluded from the calculation. The 95% prediction bands of the best fit line were calculated for the years 2010 to 2020 based on the values from 2010 to 2019. After this, the observed values for the year 2020 were added to the diagram. A statistically significant decline in the procedure of interest was assumed if the observed procedure volume for the year 2020 was below the 95% prediction band.

To identify differences between group frequencies and group distributions, Pearson’s Chi-squared test was applied (comparison of age distributions for the non-De and De groups). All statistical calculations were performed using the GraphPad Prism program v. 9.3 (GraphPad Software; San Diego, CA, USA). The statistical significance level was set at *p* = 0.05.

## 3. Results

Between 2010 and 2020, 6,308,937 orthopedic procedures were coded for patients with permanent residence in Germany and 27,420 key procedures were identified for patients with permanent residence outside Germany. A total of 13,859 procedures were performed on male patients and 13,561 on female patients in the non-DE group vs. 2,161,838 for male patients and 4,147,099 for female patients in the DE group. The proportion of female patients was significantly higher in the DE group compared to the non-DE group (ratio ♀/♂ non-DE: 1.0/1; DE: 1.9/1; x^2^ = 3208, df = 1; *p* < 0.001). The COVID-19 pandemic did not have an impact on the sex ratio between the non-DE and the DE groups (ratio ♀/♂ for the period 2010–2019 for the non-DE group: 1.0/1; and for the DE group: 1.9/1; for 2020, for the non-DE group: 0.8/1; and for the DE group: 1.7/1; x^2^ = 0.936, df = 1; *p* = 0.33). The most frequently coded orthopedic procedures in descending order turned out to be “hardware removal”, “knee arthroplasty”, and “foot surgery” for the DE and the non-DE groups, respectively (Table 1). A statistical analysis of the distribution of surgical procedures showed a significant difference between the DE and the non-DE groups (χ^2^ = 6661, df = 5, *p* < 0.001): surgical procedures, including knee arthroplasty and foot surgery, were underrepresented, whereas hardware removal and osteotomies were overrepresented in the non-DE group compared to the DE group (Table 1).

A linear regression analysis of the six different OPS codes only showed a significantly different slope between the DE and non-DE groups for the OPS code “hardware removal” (Table 2), indicating an upward trend in the DE group compared to a decreasing number of hardware removals in the non-DE group over the years.

During the outbreak of the COVID-19 pandemic, a special focus was placed on the year 2020. For this, we concentrated on the change in the total number of OPS codes, as well as the relative count (Table 3 and Table 4). The number of “bone biopsy” procedures performed in the DE group was almost at the same level in 2020 as in 2019 (aproximately 12,200 procedures per year). A moderate decrease in bone biopsies was observed among medical tourists (52 procedures in 2019 vs. 43 procedures in 2020; Figure 1 and Table 4). For the OPS codes “hardware removal” and “foot surgery”, both the DE and non-DE groups fell below the 95% prediction interval, indicating a statistically significant volume decline both for inbound and domestic patients (Figure 1 and Table 4). At the same time, the procedures coded as “knee arthroplasty” and “osteotomy” only fell below the 95% prediction interval for the non-DE group, showing a statistically significant decline only for medical tourists (Figure 1 and Table 4). A decline was also observed for the arthrodesis procedure both for the DE and non-DE groups, but only being statistically significant for the DE group.

Regarding the age distribution of domestic and inbound patients, there were statistically significant differences for patients undergoing bone biopsy, hardware removal, and arthrodesis, with inbound patients yielding an overall younger age at time of surgery (Figure 2 for relative values and Table 5 for absolute values) (χ^2^ = 29.02, df = 3, *p* < 0.01).

## 4. Discussion

Nowadays, travel and tourism are among the most valued leisure activities worldwide. With MT, a relatively new branch under the umbrella sector of travel and tourism has emerged with both economic, social, and political implications. However, a sharp definition of MT is still lacking, hampering ongoing research on this topic, often by conflating MT with health tourism and medical travel [12,13].

Traveling for the enhancement of one’s health and well-being is not a new phenomenon, as it dates back far to ancient times. The Sumerians (about 4000 BC) were probably the first to build health complexes around hot springs, attracting people from all around for the proposed therapeutic effects of thermal medicine [14,15]. The ancient Greeks set the fundamentals of health tourism and healthcare travel, attracting people from far away to the Temple of Asclepius, which in those times had transformed into a vivid health center with baths, hot springs, gymnasiums, and snake farms [15,16]. Still today, spa tourism, wellness tourism, and pilgrimage is as present as ever before, with a significant contribution to the whole health tourism sector [17]. Nevertheless, a clear distinction between MT and health tourism should be made. While health tourism usually summarizes different forms of traveling for health-related reasons, including spa and wellness tourism, offshore surgery, and dental procedures, the scope of MT is of a more confined nature [18]. Many authors have suggested that the remit of MT should include invasive procedures such as surgery, as opposed to wellness and spa tourism, where the focus is typically put on preventive care and lifestyle treatments [19,20,21]. At the same time, not all tourists that undergo invasive treatments and procedures abroad can be assigned to medical tourism, as shown by Wongkit and McKercher, who found 39.7% of surveyed visitors to Thailand that underwent medical procedures did so without the intention of receiving medical care at the time of departure from their country of residence [22]. By definition, MT should therefore include the intention of undergoing medical procedures abroad as the primary reason for traveling. Cohen has developed a fivefold topology addressing this issue: (1) “mere tourist”; (2) “medicated tourist”; (3) “medical tourist proper”; (4) “vacationing patient”; (5) “mere patient” [23]. The vacationing patient visits the country of destination with the primary intention of getting medical care, but the trip entails some vacationing activities as well. The mere patient travels with the sole intention of getting medical care and does not take part in vacationing activities. The mere patient and the vacationing patient are collectives that can be clearly assigned to medical tourism. Due to the nature of the elective orthopedic procedures analyzed in this study, patients should be assigned to the latter two (mere patient and vacationing patient).

The reasons for seeking medical care abroad are diverse and range from better medical care to easier accessibility to lower overall costs. In particular, the steep price difference of medical care between developed countries and developing countries is often cited as the driving force of patient flows from industrial states to less developed states for medical procedures. In particular, medical services are estimated to cost one fifth to one tenth in India compared to industrial countries [24]. At the same time, a patient flow from developing countries to industrial states is observed, which is often generated by wealthy people from developing countries seeking the high-quality care of high-tech medicine in countries such as the USA, Western Europe, and the UK. For Germany, this is well demonstrated by a high number of inbound Russian patients seeking medical care in the eastern part of Germany, especially in the state of Saxony [20].

However, data on the demographic characteristics of patients and the type of medical care that is typically sought after are still sparse. [25]. Moreover, treatment of the musculoskeletal system is thought to belong to the second most common treatment modalities that foreign patients engage in [10,26,27].

As a result of the conducted study, an important discrepancy regarding the age and gender distribution was found between inbound patients and domestic patients residing in Germany. Patients from abroad seeking medical treatment in Germany were overall of a younger age with a predominance of the male gender. This finding is in line with the current literature. Guy et al. also found a male predominance in a survey asking 194 American residents for their willingness to seek medical treatment abroad [28]. Noree et al. further demonstrated a male predominance among traveling patients seeking medical care in Thailand [26]. The reason for this phenomenon is still unclear and of ongoing research. However, the discrepancy between male and female patients getting proper medical care is not completely new and is more commonly known as “gender bias in medicine”, describing an overall male predominance in receiving medical care, despite a balanced distribution of diseased males and females [29]. The possible reasons for this gender inequality may be seen in hindering socioeconomical and familial circumstances in the country of origin that account more for the female than for the male population. Identical patterns of age and gender distribution for traveling medical tourists are described for host countries such as Egypt, Iran, and India [30,31,32]. Overall, for the analyzed period of 2010 to 2020, there was an upward trend for the majority of procedures evaluated, indicating a growing trend for medical tourists seeking healthcare abroad [33].

With the outbreak of the COVID-19 pandemic, an overall drop in performed procedures was seen for the domestic population in Germany, as well as for medical tourists seeking healthcare in Germany. However, the data of this study suggest that the decline in performed elective orthopedic procedures turned out to be of a higher degree for the inbound tourists seeking medical care. Depending on the elective procedure of interest, there were great differences in volume changes, with the OPS-coded “bone biopsy” yielding the least decline both for the DE and non-DE groups. Such differences and variance within the elective orthopedic procedures can be explained by the inconsistent definition and perception of elective and urgent procedures during the COVID-19 pandemic. Therefore, as bone biopsies tend to have a strong correlation with the field of cancer surgery, rendering it a more urgent procedure, it is natural that this kind of surgery was least hampered by the COVID-19 pandemic. At the same time, the overall higher decline in elective orthopedic surgery for traveling tourists can be explained by the wide national and international travel restrictions during the pandemic, hampering arrival and medical care at the country of destination. This is well in line with the data on travel and tourism during the COVID-19 pandemic, showing a 67% decline in international arrivals in Western Europe from January to December 2020 compared to data from the previous year [34]. Due to the severe restrictions on aviation activities during the pandemic, it is conceivable that the remaining medical tourists were probably from nearby countries within the European Union, especially less developed countries from Eastern European, which can be easily reached by car, train, or bus. As elective procedures had to be postponed during the pandemic, it might also be possible that the incoming medical tourists were more often considered urgent cases requiring prompt treatment.

One limitation of our study is the small number of OPS codes we evaluated (six in total). For our study, it was important to exclude patients who visited Germany as regular tourists and had to undergo an emergency surgical therapy in a German orthopedic hospital. The data provided by the Federal Statistical Office do not distinguish between emergency treatment and elective surgical therapy. For this reason, we had to carefully select the OPS codes. The six OPS codes that we analyzed here are commonly used by orthopedic surgeons in their clinical practice only for elective surgical interventions. Furthermore, we must clearly emphasize that one OPS code does not always correspond to one patient. It frequently happens that several OPS codes are coded for one patient during one operation.

## 5. Conclusions

The exact economic value of MT in the national and global context remains elusive due to inconsquent data collation and data structure. However, by the growing interest and research in the field of MT, the economic, financial, and regulatory impact of MT is thought to increase. Moreover, MT is subjected and prone to global instablities and crises, as with the onset of the COVID-19 pandemic, a greater decline in performed orthopedic procedures was observed for inbound traveling patients than for the domestic population. Therefore, intensive efforts should be initiated by German hospitals and healthcare providers to preserve the MT sector in the post-COVID-19 era.

## Figures and Tables

**Figure 1 ijerph-20-01209-f001:**
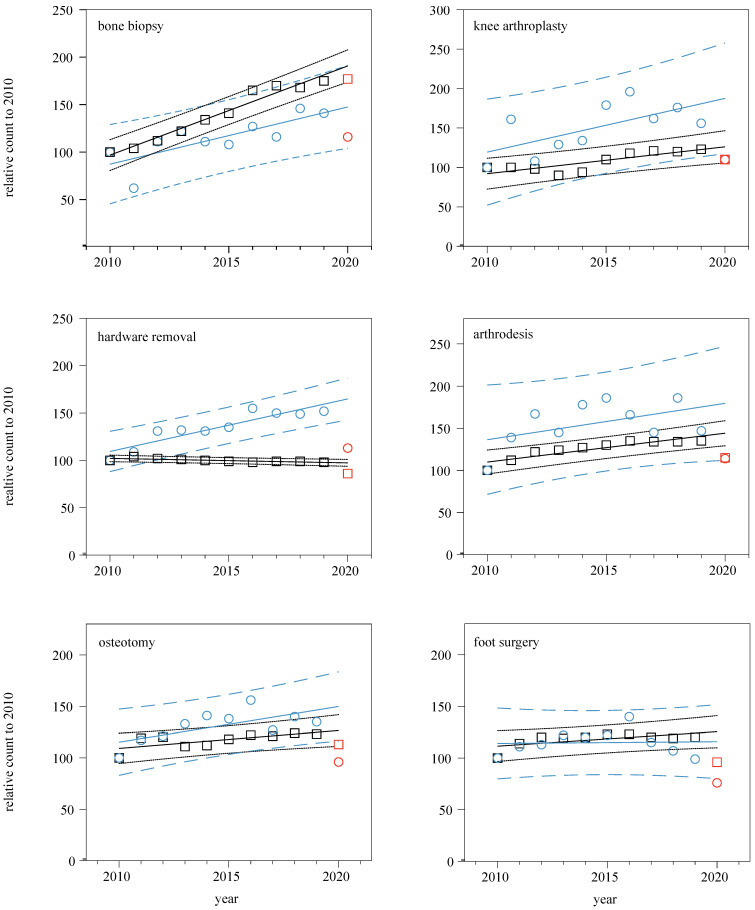
Relative change for each orthopedic procedure for the years 2010–2020. The relative count for each surgical procedure (bone biopsy, knee arthroplasty, hardware removal, arthrodesis, osteotomy, and foot surgery) was expressed as a fraction (in %) of each value (per year and group) divided by the corresponding value (per group) for the year 2010. Data for the non-DE group are given in the diagram with blue circles (“○”), and data for the DE group are marked with black squares (“▢”). The straight blue line represents the linear regression of the non-DE group, whereas the straight black line illustrates the linear regression of the DE group. The 95% prediction intervals of each dataset are given as dotted lines (small dotted line for the DE group and big dotted line for the non-DE group). The values for the year of 2020 are displayed in red color. The absolute values can be found in Table 4.

**Figure 2 ijerph-20-01209-f002:**
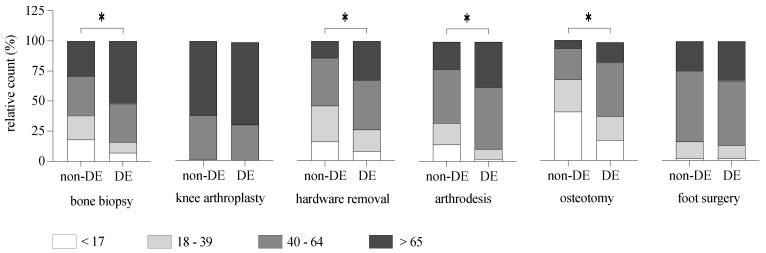
Relative count for orthopedic procedures per age group. The relative count for each surgical procedure (bone biopsy, knee arthroplasty, hardware removal, arthrodesis, osteotomy, and foot surgery) was expressed as a percentile fraction of the age subgroup (<17, 18–39, 40–64, or >65) divided by the sum of all subgroups. The absolute values can be found in Table 5. Pearson’s Chi-squared test (DE vs. non-DE; Chi-squared; * *p* < 0.01).

**Table 1 ijerph-20-01209-t001:** Number of OPS codes for both groups between 2010 and 2020.

	Number of OPS Codes	
	Non-DE	DE *
bone biopsy	466 (1.7%)	108,882 (1.7%)
knee arthroplasty	5255 (19.2%)	1,866,898 (29.6%)
hardware removal	12,453 (45.4%)	1,921,896 (30.5%)
arthrodesis	1707 (6.2%)	450,175 (7.1%)
osteotomy	3441 (12.5%)	348,813 (5.5%)
foot surgery	4098 (14.9%)	1,612,273 (25.6%)
Sum	27,420 (100%)	6,308,937 (100%)

Total number for six OPS codes (bone biopsy, knee arthroplasty, hardware removal, arthrodesis, osteotomy, and foot surgery) for the years 2010–2020 subdivided for the DE and the non-DE groups, as well as the percentage of the sum (in parenthesis). Χ^2^ = 6661, df = 5, * *p*< 0.001 vs. non-De.

**Table 2 ijerph-20-01209-t002:** Results of the linear regression analysis for both groups (DE and non-DE).

OPS Code	Slope	Intercept	R^2^	Is Each Slope Significantly Non-Zero?	Are the Differences between the Slopes (DE vs. Non-DE) Significant?
Bone biopsy					
non-DE	6.024	−12,021	0.6058	yes (*p* = 0.0080)	no (F = 3.327; *p* = 0.0869)
DE	9.388	−18,773	0.9609	yes (*p* < 0.0001)	
knee arthroplasty					
non-DE	6.818	−13,585	0.0433	yes (*p* = 0.0392)	no (F = 1.400; *p* = 0.2540)
DE	3.406	−6754	0.6904	yes (*p* = 0.0029)	
hardware removal					
non-DE	5.552	−11,049	0.8341	yes (*p* = 0.0002)	yes (F = 46.12; *p* < 0.0001)
DE	−0.4727	1052	0.5761	yes (*p* = 0.0109)	
arthrodesis					
non-DE	4.321	−8549	0.2464	no (*p* = 0.1445)	no (F = 0.1074; *p* = 0.7472)
DE	3.424	−6773	0.8101	yes (*p* = 0.0004)	
osteotomy					
non-DE	3.467	−6853	0.4617	yes (*p* = 0.0307)	no (F = 1.418; *p* = 0.2511)
DE	1.733	−3375	0.5059	yes (*p* = 0.0211)	
foot surgery					
non-DE	0.1758	−239	0.0019	no (*p* = 0.9038)	no (F = 0.6288; *p* = 0.4394)
DE	1.394	−2690	0.3933	no (*p* = 0.1092)	

Linear regression analysis for the “relative count to 2010” for the period 2010–2019 for the DE and the non-DE groups (complementary table to Figure 1). Values are given for the non-DE and the DE groups (slope, intercept on the y-axis, R^2^, as well as level of significance) for simple regression lines.

**Table 3 ijerph-20-01209-t003:** Comparison of relative procedure counts for the years 2010–2019 and 2020.

Procedure (Relative Count)	Mean Relative Count from 2010 to 2019	Mean Increase per Year for 2010 to 2019	Observed Relative Count 2020
bone biopsy			
non-DE	114	7.7%	116
DE	139	6.5%	177 *
knee arthroplasty			
non-DE	150	8.3%	110 *
DE	107	2.5%	110
hardware removal			
non-DE	134	5.0%	113 *
DE	100	−0.2%	86 *
arthrodesis			
non-DE	156	6.4%	114 *
DE	125	3.5%	115 *
osteotomy			
non-DE	131	3.9%	96 *
DE	117	2.6%	113
foot surgery			
non-DE	114	0.4%	76 *
DE	117	2.1%	96 *

Comparison of the mean relative counts and the relative increases per year of the six different OPS codes for the years 2010 to 2019 with the observed relative count of the pandemic year of 2020. Significant differences between the observed relative count 2020 vs. the mean relative counts from 2010 to 2019 (* *p* < 0.05; *t*-test).

**Table 4 ijerph-20-01209-t004:** Total count for surgical procedures for the years 2010–2020.

	Year	Bone Biopsy	Knee Arthroplasty	Hardware Removal	Arthrodesis	Osteotomy	Foot Surgery
non-DE	2010	37	326	855	102	245	334
	2011	23	525	936	142	287	372
	2012	41	352	1118	170	297	379
	2013	45	421	1127	148	326	409
	2014	41	437	1119	182	346	402
	2015	40	585	1154	190	339	408
	2016	47	638	1326	169	382	469
	2017	43	528	1283	148	310	384
	2018	54	575	1273	190	342	357
	2019	52	509	1297	150	331	330
	2020	43	359	965	116	236	254
DE	2010	6941	157,570	176,993	32,886	27,177	126,636
	2011	7208	157,557	183,565	36,897	32,315	143,802
	2012	7794	154,377	180,764	40,186	32,686	152,312
	2013	8465	142,545	178,800	40,887	30,274	150,668
	2014	9314	148,599	176,529	41,830	30,441	152,108
	2015	9795	172,664	174,545	42,749	31,942	155,414
	2016	11,477	186,627	173,977	44,278	33,080	155,446
	2017	11,767	190,697	174,924	44,194	32,989	152,163
	2018	11,659	189,801	175,440	43,947	33,695	150,448
	2019	12,150	193,216	173,681	44,370	33,518	151,693
	2020	12,312	173,245	152,678	37,951	30,696	121,583

Numerical representation of the total count of the six different OPS codes for the years 2010 to 2020.

**Table 5 ijerph-20-01209-t005:** Total count for orthopedic procedures per age group.

	Age (Years)	Bone Biopsy	Knee Arthroplasty	Hardware Removal	Arthrodesis	Osteotomy	Foot Surgery
non-DE	<17	94	33	2220	305	1576	93
	18–39	112	73	3768	285	926	604
	40–64	156	2082	4987	745	741	2453
	>65	104	3067	1478	372	198	948
DE	<17	6515	417	153,764	9139	58,978	27,753
	18–39	8790	5124	331,580	35,552	72,837	191,088
	40–64	34,524	610,615	786,542	235,130	157,583	886,848
	>65	59,053	1,250,742	650,010	170,354	59,415	506,584

Numerical representation of the total count of the six different orthopedic procedures per age group.

## Data Availability

The raw data we used for the current data analysis can be found in the open repository github (https://github.com/ioannis-stratos/G-DRG_1 (accesed on 18 September 2022)).

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
