# Peer review of "Demographics and Trends for Inbound Medical Tourism in Germany for Orthopedic Patients before and during the COVID-19 Pandemic"

_ijerph, 2023, doi:10.3390/ijerph20021209_

Round 1

Reviewer 1 Report

Dear Authors:

This paper is very focused in the analisys (like a case of study) but not in the academic merit.

Plase provide justification why this issue in important, which is the contribution, previous research advance, relevancia for other European countries, etc...

In addition more information in tables are required in order to see the real data (not just figures). Descriptive analysis, evolution data, etc..

Conclusions and results should be rewritting. They are clearly insuficient.

Author Response

Dear Reviewer, 

We thank you four your valued comments on the manuscript and your expertise in this field. We think that the script has improved a lot by implementing and working on your suggestions. Below you can find a point-by-point response to your original comments: 

  1. Please provide justification why this issue in important, which is the contribution, previous research advance, relevancia for other European countries, etc...
    • We thank the reviewer for this comment. We have now implemented a paragraph in the introduction section clarifying the importance of Medical Tourism (MT). Patients are increasingly transforming into informed consumers who choose their medical provides out of a broad market that is not limited to geographical borders. The economic and financial revenue generated by MT is substantial. The European Union and especially Germany are popular destinations for medical travelers. It is therefore important to analyze incoming patient flows and the kind of medical service that is sought after. A detailed background information and data on MT are necessary in order to establish new frameworks for MT. 
    • Please refer to lines 50 to 70 of the revised manuscript.

2. In addition more information in tables are required in order to see the real data (not just figures). Descriptive analysis, evolution data, etc..

    • We appreciate your interest in the data analysis. We have revised the descriptive analysis and added extra tables showing the absolute values. 
    • Please refer to Table 4 and Table 5. 

3. Conclusions and results should be rewritting. They are clearly insuficient.

    • We thank the reviewer for the extensive review of the manuscript. We have now revised the conclusion section. Please refer to lines 333 to 340 in the revised manuscript. 
    • The results section has also been edited and some extra Table have been implemented. Please refer to lines 141 to 331 in the manuscript. Please refer to Table 4 and Table 5.

Reviewer 2 Report

First of all, I would like to greet authors for a very interesting paper with an important subject for tourism industry.

As stated by authors, there is no doubt that medical tourism is a rapidly growing sector of economic growth and diversification and therefore, this needs to be better studied.

To this end, authors have defined very well their goals.

In more detail, this paper common demographic properties and characteristics of the inbound medical tourists seeking orthopedic medical care in Germany between the years 2010 to 2020 and compare them to a domestic group.

Simultaneously, they have examined how the COVID-19 pandemic outbreak of 2020 changed the field of medical tourism in Germany.

Methodologically, calculations were performed using administrative hospital data provided by the Federal Statistical Department of Germany from the years 2010 to 2020. Six elective orthopedic surgery codes (bone biopsy, knee arthroplasty, foot surgery, osteotomy, hardware removal and arthrodesis) were identified as key service-indicators for medical tourism and further analyzed.

This turned possible to achieve to the main discussion and conclusions.

Regardless, the quality of the performed case study and the whole document, this paper doesn´t have any section about Literature Review. This is extremely important in order to allow readers to understand the theoretical discussion under this issue.

So, I recommend that authors may deepen this section even though this is extremely summarized in the introduction.

Another issue is that the paper only has 18 references. Although quantity is not quality, some major more recent papers could and should be used. So, I also recommend authors to look for some more recent references on the subject.

In the conclusions, authors have concluded “Medical tourism is a growing sector of relevant volume with potential influence on 256 the national and worldwide healthcare system and economy”. However, in any place have we seen statistics about both volume, receipts or any other kind od indicator that can help readers to understand this possibility.

Author Response

Dear reviewer,

we would like to thank you for your extensive and valued review of the manuscript. Below you can find a point-by-point response to your original comments. We feel happy to inform you that we have fully addressed all your comments, thereby improving a lot on the quality of this research. 

1.First of all, I would like to greet authors for a very interesting paper with an important subject for tourism industry.As stated by authors, there is no doubt that medical tourism is a rapidly growing sector of economic growth and diversification and therefore, this needs to be better studied.To this end, authors have defined very well their goals.

In more detail, this paper common demographic properties and characteristics of the inbound medical tourists seeking orthopedic medical care in Germany between the years 2010 to 2020 and compare them to a domestic group.

Simultaneously, they have examined how the COVID-19 pandemic outbreak of 2020 changed the field of medical tourism in Germany.

      • We thank the reviewer for the extensive review of the manuscript. No changes have been applied.

2. Methodologically, calculations were performed using administrative hospital data provided by the Federal Statistical Department of Germany from the years 2010 to 2020. Six elective orthopedic surgery codes (bone biopsy, knee arthroplasty, foot surgery, osteotomy, hardware removal and arthrodesis) were identified as key service-indicators for medical tourism and further analyzed.

This turned possible to achieve to the main discussion and conclusions.

      • Many thanks. No changes so far.

3. Regardless, the quality of the performed case study and the whole document, this paper doesn´t have any section about Literature Review. This is extremely important in order to allow readers to understand the theoretical discussion under this issue.So, I recommend that authors may deepen this section even though this is extremely summarized in the introduction. 

      • Many thanks for this highly appreciated comment. We totally agree that the background information on this topic has not been sufficiently covered in the first draft of the manuscript. We have therefore revised the discussion section of the manuscript including a short historical overview of medical tourism and the problematic of its definition. We have further reviewed the different reasons that might drive cross border health care. 
      • Please refer to the discussion section (line 233 - 331) in the revised manuscript. 
      • We have also edited the introduction section and added some more background information regarding medical tourism in the European Union. Please refer to line 50 - 70. 

4. Another issue is that the paper only has 18 references. Although quantity is not quality, some major more recent papers could and should be used. So, I also recommend authors to look for some more recent references on the subject.

      • Many thanks for this important note. With an extensive literature review, we have now updated the reference list and nearly doubled the reference number to 34. Please refer to the reference list in the revised manuscript. 

5. In the conclusions, authors have concluded “Medical tourism is a growing sector of relevant volume with potential influence on 256 the national and worldwide healthcare system and economy”. However, in any place have we seen statistics about both volume, receipts or any other kind od indicator that can help readers to understand this possibility.

      • Many thanks. We have now included data on the economic and financial value of medical tourism in the European Union and in Germany. Unfortunately, due to patchy and incomplete data aggregation and data structure exact values for medical tourism are elusive. We have therefore rewritten the conclusion section. 
      • Please refer to line 50 to 62 in the revised manuscript. 
      • Please refer to the conclusion section of the revised manuscript. 

Round 2

Reviewer 2 Report

Thank you for your review.

The text that was already good, is now much better with the improvements you made.

Congratulations.